# A Tough and Mildew-Proof Soybean-Based Adhesive Inspired by Mussel and Algae

**DOI:** 10.3390/polym12040756

**Published:** 2020-03-31

**Authors:** Yue Bai, Xiaorong Liu, Sheldon Q. Shi, Jianzhang Li

**Affiliations:** 1Beijing Advanced Innovation Center for Tree Breeding by Molecular Design, Beijing Forestry University, Beijing 100083, China; byueer@126.com (Y.B.); happyrong1993@bjfu.edu.cn (X.L.); 2Key Laboratory of Wood Materials Science and Utilization, Beijing Forestry University, Beijing 100083, China; sheldon.shi@unt.edu; 3College of Materials Science and Technology, Beijing Forestry University, Beijing 100083, China; 4Department of Mechanical and Energy Engineering, University of North Texas, Denton, TX 76203, USA

**Keywords:** protein adhesive, soybean meal, mold resistance, water resistance, metal complexation

## Abstract

Despite the recent advances in protein-based adhesives, achieving strong adhesion and mold resistance in wet environment is challenging. Herein, a facile fabrication technology of preparing tough bio-adhesive by incorporating soybean meal and blood meal is presented. Inspired by the marine mussel byssi and brown algae, metal coordination was introduced into a loosely bound protein system to construct multiple chemical cross-linking networks. Mixed alkali-modified blood meal (mBM) was mixed with soybean meal, then 1,6-hexane dioldiglycidyl ether (HDE) and zinc ion were introduced to fabricate soybean meal and blood meal-based adhesives. The attained adhesives exhibited good thermal stability, water resistance (the wet shear strength is 1.1 MPa), and mold resistance, with appropriate solid content (34.3%) and relatively low moisture uptake (11.9%). These outstanding performances would be attributed to the reaction of 1,6-hexane dioldiglycidyl ether with protein to form a preliminary cross-linking network; subsequently, the coordination of zinc ions with amino or carboxyl strengthened and toughened the adhesive. Finally, the calcium ions gelled the adhesives, providing cohesion force and making the network structure more compact. This study realized the value-added utilization of protein co-products and developed a new eco-friendly bio-based adhesive.

## 1. Introduction

The renewable and degradable bio-based materials generated from agricultural industry have received increasing attention because of the depleting petroleum resources and the requirement of environmental protection. For example, soy protein, rapeseed protein, wheat gluten, and corn starch have been used in biomedical application, packaging, and food engineering [1,2,3,4]. Notably, proteins exhibit high cost-effectiveness, are non-specific, and are abundant resources, which are promising bio-based materials to replace petroleum-based polymers [5,6,7].

Soybean protein adhesive, as a renewable bio-based product, has shown great potential because of its low price and is environment friendly since 1930s. However, the low bonding strength, poor water and mildew resistance of the protein-based adhesives have limited their applications. Up to date, some modification methods have been effective in improving the performances of soybean protein adhesive, such as denaturing agent- [8], cross-linking agents- [9,10], and biomimetic chemistry modification [11], etc. However, all of these methods involved in using the fossil resources, and the modification process is cumbersome, which have increased the cost and processing time for the adhesive preparation. 

Biomass material blending is a simple, green, and elegant way to improve the performance of composite materials. Proteins hybridization has been proved to be an effective way to improve the performance of protein-based adhesives and films because of their similar inherent structure and satisfactory biocompatibility [12,13,14].

Blood meal, a low-value slaughterhouse byproduct, with a high protein content (about 80%), exhibits better bonding strength and water resistance compared with other proteins and provides the composites with a higher mechanical property [15,16]. However, the composition of blood meal is non-homogeneous, and shows poor solubility, leading to its very limited processability [17,18]. Organic solid wastes, such as blood meal, have been banned from being used as animal feed or even food crop fertilizer in some countries because of the potential high concentration of toxic elements [18,19]. Therefore, it is desirable to realize an appropriate way to utilize the blood meal for value-added products with a reduction of wastes.

In nature, many organisms, such as benthic algae and marine mussels, have evolved their own strategies for adhering to a variety of wet and even underwater surfaces [20,21]. In adhesive mussel foot proteins, the selective metal ion contents (e.g., copper, iron, manganese, and zinc) were four, five, and up to 100,000 times more than that of open ocean waters [17,22]. Metal ions and metal ion chelating proteins have shown to play an important role in the adhesion process of mussels [23]. Unlike the mussel, the bulk of brown algae adhesive is a separate network, with phenolic and alginate groups, gelled by calcium ions to provide cohesion, and then crosslinked together to form the adhesive [24]. Metal coordination is a unique chemical interaction. The coordination interaction between large biomacromolecules (especially proteins) and common transition metal ions (such as iron, zinc, copper and nickel) can give materials higher adhesion, toughness, and hardness [25]. The introduction of soluble salts to construct metal ion complexation has proved to improve protein-based adhesive working properties and adhesive performance [26].

In this work, a simple way is proposed to synergistically combine two strategies of mussel and algae to prepare the protein-based adhesives. The multiple network structure crosslinked by metal ions was designed to provide the adhesive with excellent water resistance. The addition of blood meal as the enhancement phase not only increased the protein content of the system, but also provided more active sites for metal complexation. The 1,6-hexane dioldiglycidyl ether was introduced to bestow good mildew resistance of the adhesive and denser crosslinking structure. It is believed that the protein-based adhesive with toughness, good water resistance, and mold resistance could be obtained by combining soluble soybean meal and insoluble blood meal, and introducing metal complexation and chemical cross-linking.

## 2. Experimental

### 2.1. Materials

Soluble soybean meal (53% protein, 200 mesh) was purchased from Xiangchi Grain and Oil Co., Ltd. (Shandong, China). Blood meal (approximately 80% protein, 200 mesh) was purchased from Wei Duofeng Biotechnology Co., Ltd. (Shandong, China). Zinc chloride (AR, 98%) was purchased from Shanghai Macklin Biochemical Co., Ltd. (Shanghai, China). 1,6-Hexane dioldiglycidyl ether, sodium hydroxide, calcium hydroxide, and sodium silicate were purchased from Beijing Chemical Reagents Co., Ltd. (Beijing, China). Poplar veneers (400 × 400 × 1.6 mm^3^, 8.0% moisture content) were purchased from Wen’an Plywood, Ltd. (Hebei, China).

### 2.2. Preparation and Characterization of Soybean Meal and Blood Meal-Based Adhesive

(1) Preparation of the Modified Suspension of Blood Meal

The preparation of the modified suspension of blood meal was in accordance with the procedure described in the previous report with some modifications [27]. Briefly, blood meal (10 g) was suspended in tap water (25 g) for 2 h, and then lime milk (the ratio of calcium hydroxide to water is 1:4, 2 g), sodium hydroxide (30 wt %, 1 g), and sodium silicate (1.5 g) were added successively every 1 min at room temperature.

(2) Preparation of the ZnCl_2_ Modified Soybean Meal and Blood Meal-Based Adhesive

The adhesives were prepared according to the method of Zhang et al. [28], as follows. The soybean meal adhesive (adhesive I) was prepared by 30-min stirring of a mixture of soybean meal (25 g) and water (50 g) and HDE (2.25 g) at room temperature to form a homogeneous system. The ZnCl_2_-modified soybean meal-based adhesives (adhesive II) were fabricated by 30-min stirring of a solution of soybean meal (25 g) and HDE (2.25 g) in water (50 g) followed by the addition of ZnCl_2_. The soybean meal and blood meal-based adhesives (adhesive III) were prepared by dissolving a mixture of soybean meal (15 g), the modified suspension of blood meal (39.5 g), and HDE (2.25 g) in water (25 g). Subsequently, 0.75, 1.125, and 1.5 g ZnCl_2_ were added, respectively and stirred for 30 min. The formulations of all the adhesives are summarized in Table 1.

### 2.3. Preparation of Three-Layered Plywood

Poplar veneer was cut into 200 × 200 × 1.6 mm^3^ size pieces. Three-layer plywood was prepared as follows: about 180 g/m^2^ of adhesive was coated on each layer. Then, the plywood was hot-pressed under the conditions of 120 °C and 1.0 MPa for 315 s. The as-prepared plywood samples were stored in the ambient condition for 24 h before testing.

### 2.4. Characterization of Soybean Meal and Blood Meal-Based Adhesive

The adhesive samples were dried in an oven at 120 ± 2 °C until a constant weight and then ground into 200-mesh powder for use.

ATR-FTIR analysis was carried out on a Nicolet 6700 spectrometer (Thermo Fisher Scientific Inc., Madison, WI, USA) over a region of 500−4000 cm^−1^ using 32 scans at a resolution of 4 cm^−1^ under the ambient temperature. The thermal stabilities of cured adhesives were measured on a TGA instrument (Q50, WATERS Company, Milford, MA, USA). Approximately 7 mg of powdered adhesive samples were weighed and scanned from 30 to 610 °C at a heating rate of 10 °C/min in a nitrogen environment. DSC (Q2000, TA instrument, Newark, DE, USA) was applied at heating and cooling rate of 10 °C/min and a heating range between −40 to 150 °C. Scanning electron microscopy (SEM, A Hitachi S-3400N, Ibaraki, Japan) was used to examine the cross-sectional surfaces of adhesive samples. The X-ray diffraction (XRD) patterns were recorded by an X-ray diffractometer (Bruker D8, Karlsruhe, Germany) using Cu Kα radiation (40 kV, 40 mA) with a 2θ range between 5°and 60°.

### 2.5. Adhesive Performance Test

(1) Toughness Measurement

The toughness of cured adhesives was measured by crack observation. To be specific, the adhesive samples were coated on the glass sheet, placed in the oven at 120 ± 2 °C for 30 min, and then cooled to room temperature. The cooled adhesives were photographed using a digital single-lens reflex camera to observe the cracks.

(2) Solid Content Measurement

Approximately 2 g of the samples were weighed, dried in a 120 ± 2 °C oven for 3 h, and weighed after cooling at room temperature for 15 min. The solid content was calculated using Equation (1):(1)$$\mathrm{Solid}\;\mathrm{content}\;(\%)=\frac{m_{2}-m_{0}}{m_{1}-m_{0}}\times 100$$
where $m_{0\;}$ is the weight of the aluminum foil, $m_{1}$ and $m_{2}$ are the weights of the adhesive samples before and after drying, respectively. 

(3) Moisture Uptake Measurement

The cured adhesives were placed in an environmental chamber with a temperature of 50 ± 2 °C and a relative humidity of 80 ± 2% relative humidity. The weight of the adhesive samples were recorded every 2 h until a constant weight was reached. The moisture uptake was then calculated using Equation (2): (2)$$\mathrm{Moisture}\;\mathrm{uptake}\;(\%)=\frac{m_{4}-m_{3}}{\;m_{3}}\times 100$$
where $m_{3}$ and $m_{4\;}$ are the weights of the adhesive before and after moisture uptake, respectively.

(4) Water Resistance

The plywood was cut into 25 mm × 100 mm rectangular specimens according to the China National Standard GB/T17657−2013. The rectangular specimens were immersed in water at a temperature measured as 63 ± 2 °C for 3 h and then cooled at room temperature for 10 min prior to measuring the wet bonding strength. The wet shear strength of the adhesive bonded plywood was measured using a universal testing machine (WDW-200E, Jinan, China) at a cross-head speed of 10 mm·min^−1^. The wet shear strength was calculated using Equation (3): (3)$$\mathrm{Wet}\;\mathrm{shear}\;\mathrm{strength}\;(\mathrm{MPa})=\frac{\;\mathrm{Force}\;\left(N\right)}{\mathrm{Gluing}\;\mathrm{area}\;\left({\mathrm{mm}}^{2}\right)}$$

(5) Mold Resistance

The adhesive samples were mildewed in a chamber with a constant temperature at 30 ± 2 °C and a relative humidity of 90 ± 2%, and the moldy conditions were observed every 24 h. 

## 3. Results and Discussion 

### 3.1. Design, Synthesis, and Characterization of Soybean Meal and Blood Meal-Based Adhesive 

To integrate the multiple functionalities into the adhesive, including high bonding strength, water resistance and anti-mycotic capability, a facile but delicate strategy was designed as illustrated in Scheme 1. At first, in an alkaline environment, blood meal was formed at a uniform dispersion. The alkali treatment hydrolyzed the peptide or amide bonds, which resulted in the transformation of the long chain polypeptide into low molecular weight products, and the polar and hydrophobic groups were exposed on the surface of the protein [29]. Besides, sodium silicate not only provided an alkaline environment, but acted as a curing agent [18]. The incorporation of metal ions with a group capable of complexing increased the density of the crosslinked network and led to a remarkable increase in strength.

As shown in Figure 1, the ATR-FTIR spectra of the adhesives shows the changes in functional groups and possible interactions in the mixed protein bio-adhesive system. The broad absorption band at 3271 cm^−1^ was attributed to the free and bound O–H and N–H bending vibrations that could form hydrogen bonds with carbonyl groups in the peptide linkages [29]. The typical amide bands in the soy protein adhesives were found at 1644, 1515, and 1233 cm^−1^ corresponding to amide I (C–O stretching), amide II (N–H bending), and amide III (C–N and N–H stretching) respectively [28]. For the formulation 1, the peak at 3271 cm^−1^ shifted toward 3274 cm^−1^, and the abroad absorption band region at 3100–3600 cm^−1^ became wider, mainly because the HDE reacted with the amino groups of soybean protein during the curing process to form a covalent crosslink network, as shown in Scheme 1b. The peaks observed at 2929 and 2869 cm^−1^ were due to the symmetric and asymmetric stretching vibrations of the –CH_2_ groups [30,31]. When using soybean meal and blood meal as the matrix, it was seen that the peak at 2869 cm^−1^ was slightly shifted to 2865 cm^−1^, revealing an interaction occurred between protein and zinc ions [32]. The peak at 1515 cm^−1^ (amide II) obviously shifted to 1528 cm^−1^, the intensity of the peak at 1233 cm^−1^ reduced in comparison to the formulation 1 and a new peak appeared at 1062 cm^−1^. These results indicated that a complex physical/chemical reaction took place between Ca^2+^ or Zn^2+^ and protein.

Figure 2 shows the X-ray diffraction patterns and the corresponding crystallinity used to identify the multiple crosslinking interactions in the protein adhesive. The characteristic peaks at 2θ of approximately 9°and 19.6° represented the α-helix and β-sheet structures of protein secondary conformations, respectively [33]. After the incorporation of Ca^2+^ or Zn^2+^, the peak at approximately 9° almost disappeared and the peak at 19.6°showed obvious decrement and faintly shifted to a lower degree, which revealed that the multiple interactions effectively disrupted the protein conformation after forming coordination bond and intermolecular H-bonds. Crystallization is the ordered array of the peptide chains’ arrangement and the decrease of crystallinity means the increase of crosslinking [34]. The results of this experiment showed that crystallinity of adhesive samples were in the order adhesive I (formulation 1) > adhesive II (formulation 3) > adhesive III (formulation 6) (Figure 2). First of all, the chemical (e.g., alkali, sodium silicate) treatment destroyed the structure of protein, which was displayed by decreased crystallinity, releasing more functional groups for cross-linking reactions. Therefore, for the adhesive III, the decrease of crystallinity revealed the formation of multiple cross-linking interactions between active hydrophilic groups and Ca^2+^ or Zn^2+^, which increased the crosslinking density of the system. 

It is known that the thermal stability of adhesive is of great importance to water resistance property of the resulted product, and hot-pressing process. It could be seen that the thermal degradation of soybean meal and blood meal-based adhesives was divided into three stages from Figure 3a. The thermal degradation data is listed in Table 2. In the stage of 60 to 150 °C (stage I), the thermal weightlessness rate (M**_S_****_I_**) was relatively low (≤5%), which was mainly caused by releasing of moisture in the adhesive, while there was no protein degradation [35]. In the stage II, the temperature was between 150 and 260 °C, the degradation behavior was mainly attributed to the unreacted micromolecular decomposition and some unstable chemical bond cleavage, including intra-/intermolecular hydrogen bonding, electrostatic bonds, and cleavage of the covalent bonding between the peptide bonds [36,37]. The degradation of protein skeleton occurred in the stage III (260–600 °C). In this stage, peptide bonds in the primary structure of proteins and C−O, S−S, and O−N bonds that stabilized the tertiary structure broke, thereby creating a large mass loss [9]. During the stages II and stages III, the rate of decomposition increased in the formulation 1 to a greater extent than in formulation 6 (Figure 3b), which could be reasonably ascribed to the breakage of intramolecular and intermolecular hydrogen bonds of protein molecules at higher pH [38]. After incorporating the modified blood meal in the adhesive, the weight loss was reduced to 46.3% (formulation 7), suggesting that blood meal and soybean protein had a good co-effect and the chemical interaction among crosslinkers, ions, and protein formed a compact crosslinking network to prevent the gas diffusion and decomposition of the matrix. In addition, it was found that the DTG peak (Tmax) in the protein backbone degradation stage increased from 301 °C (formulation 1) to 308 °C (formulation 6) and exhibited a higher residue weight compared to the control (Table 2). Therefore, the multiple interactions in the system could effectively strengthen the structure and prevent the breakage and movement of protein chains, thereby improving the thermal stability of the adhesive. 

The thermodynamic properties of pristine soybean meal and modified soybean meal-based adhesives were further studied by DSC analysis. The pristine soybean meal and blood meal showed two endothermic transitions that were caused by the denaturation of globulins (Appendix A). However, there only existed a single glass transition on the DSC curves of adhesive I (formulation 1), adhesive II (formulation 3), and adhesive III (formulation 6) (Figure 4), indicating that the phases in the system had excellent miscibility and it also supported molecular interaction of macromolecules and co-cross-linking between soybean meal and blood meal [39]. In addition, the adhesive III exhibited the highest glass-transition temperature (*T*_g_ = 63 °C) among the fabricated adhesives and therefore had the lowest chain mobility. This was because complex chemical crosslinked structure severely restricted the movement of the protein segments. 

Scanning electron microscopy (SEM) was used to investigate the morphology of the fracture surface of the adhesives. As shown in Figure 5, a number of holes caused by the evaporation of moisture, swelling to break the bond during the hot-pressing process were observed on the fracture surface of the formulation 1, indicating that the structure of soybean meal adhesive was loose and moisture was easy to move in and out. Therefore, the water resistance and the bond strength of the plywood were relatively low. Compared with that without Zn^2+^, the adhesive II (formulation 2, 3, 4) showed ductile fracture, which is consistent with the observation of surface crack [40]. Additionally, the fracture surface of the Zn-modified soybean meal-based adhesives become rougher, meaning that the formation of metal-ligand interactions led to tighter crosslinking. However, there were still some holes on the fracture surface of the formulation 3. From the cross-sectional images of formulation 6, it was shown that the inclusion of the modified blood meal into the adhesive could improve the density of adhesive layer structure on account of more metal binding sites complexed with the corresponding ions to form a multiple crosslinked network. Moreover, it also helped to form a strong mechanical interlock between the adhesive and the wood, so as to improve the bonding strength. However, when the amount of zinc ions increased to 2% (formulation 7), it was found that there was obvious aggregation phenomenon as marked by the red circle. Excessive ions would precipitate the protein and partially separate out, resulting in stress concentration and reduced water resistance. 

### 3.2. Performances of the Soybean Meal and Blood Meal-Based Adhesive

The solid content, moisture uptake, and the crack of the soybean meal and blood meal-based adhesive were tested to evaluate the properties of the adhesive. The solid content of adhesive has an important effect on the bonding quality and the energy consumption of hot pressing. High solid content will make the coating difficult, while low solid content will increase the energy consumption and influence its performance in bonding with the plywood. Protein-based adhesives have better bonding properties when the solid content is between 32% and 36% [41]. As shown in Table 3, the solid contents of zinc ion-added adhesives were over 32%, implying that the ionic-modified protein adhesive had good bonding quality. Moisture uptake reflects the stability of cured adhesives. Theoretically, after the mixed alkali treatment, the protein molecular chain stretched and more hydrophilic groups (e.g., –OH, –COOH) were exposed, which would increase the moisture uptake of the adhesive [42]. However, the moisture absorption was significantly reduced after the introduction of mBM. In particular, formulation 6 achieved a minimum hygroscopicity of 11.8%, which indicated that blood meal played an important role in the hygroscopic stability of adhesive and it was reasonable to explain that the metal ions could initiate cross-linking of internal groups of proteins. Crack observation could be used to evaluate the toughness of water-based thermosetting resins. The corresponding behaviors of adhesive samples are presented in Figure 6. A large number of holes and small cracks were observed for the formulation 1, showing the brittle behavior. The addition of zinc ions reduced the brittleness with almost no holes and cracks that could be seen for formulation 2, 3, 4, 5, and 6 (Figure 6a). This suggested that the interactions between soybean meal, blood meal, and Zn had a plasticizing effect on protein matrix and improved the toughness of the material [43]. It was noticed that when the content of zinc ions increased to 2%, large and wide cracks appeared on the film (Figure 6a, formulation 7), which was due to the excessive metal ions leading to protein precipitation and salting out. This phenomenon could also be observed directly by SEM of the cross-section morphology of the adhesive (Figure 6b). Therefore, the compatibility of the metal ion content with the adhesive must be considered. More metal particles will lead to the failure of adhesive formation and/or complete loss of adhesiveness because of particle packing defects. 

### 3.3. Water Resistance 

The wet shear strength of plywood fabricated using the soybean meal, blood meal-based adhesives are presented in Figure 7. In particular, the wet shear strength of plywood bonded by the formulation 1 was 0.48 MPa, which failed to meet the type II plywood requirement (≥0.7 MPa) of the China National Standards (GB/T17657−2013). It was found that the incorporation of zinc ions into soybean meal-based adhesive resulted in much improved wet shear strength and the adhesive II had a wet shear strength greater than 0.7 MPa (i.e., formulation 2–0.87 MPa, formulation 3–0.78 MPa, formulation 4–0.77 MPa), indicating that the zinc ions formed chemical crosslinking with the protein, which improved the water resistance of the adhesive. Therefore, on the basis of the metal-protein cross-linking reactions, blood meal was blended into the system to improve the protein content and further increased the cross-linking sites. However, the simple addition of blood meal did not seem to have a significant effect to water resistance of plywood (Appendix A). When the blood meal was modified by mixed alkali, the shear strength of the resulted plywood was significantly improved, reaching 1.1 MPa (i.e., formulation 6), which was 41.0% higher compared with the formulation 4. This was mainly because that the alkali treatment could activate the blood meal, the hydrophilic groups were exposed to form insoluble substances by complexation with calcium or other ions [44]. As such, when subjected to external forces, the multi-crosslinked network structure in the system can effectively resist the damage of mechanical load and ensure strong cohesive interactions between protein molecules. In summary, the obtained excellent water resistance of adhesive III is based on the following three reasons: First, soybean protein and blood protein as an organic component provided more binding sites for chelation of inorganic metal ions [45]. To be specific, the zinc ions coordinated with the amino group of the protein to form an insoluble coordination compound, and the chain was linked by an ion or a coordinate bond [25], which primarily increased the crosslink density, making the structure of the adhesive more compact and prevented swelling by water. Second, the presence of calcium ions enhanced water resistance of the protein-based adhesives because of which the calcium ions might chelate with the carboxyl group of the protein. Then there was the slow conversion of the sodium salt of polypeptide chains to water insoluble calcium salts [46]. Finally, the mechanism of metal-ion-mediated hardening improved the mechanical properties of protein matrix. This was well documented in both the spider fang and the mussel byssus cuticle [47,48].

### 3.4. Mold Resistance

The sensitivity of protein-based adhesives to microbial erosion is a major factor restricting their industrial application. The mold resistance of the modified adhesives was evaluated by observing the growth of mold on the surface of the adhesive. The adhesive I and adhesive III (formulation 6) exhibited obvious antimycotic properties, whereas the pristine soybean meal-based adhesive failed to inhibit the growth of mold (Figure 8). After the mildew treatment for 24 h, a large amount of colonies appeared on the surface of the soybean meal adhesive (SM), over time, the density of the colonies gradually increased, and the color changed from the original milky white to gray-brown. However, only 3% of HDE could give the system better mildew resistance. The colonies did not appear on the surface of the adhesive I and adhesive III until the 6th day and 14th day, respectively. Although there have been no reports on the use of HDE in soybean protein adhesives to prevent mold, this study proved that HDE did have an effect on the mildew resistance of protein-based adhesive. In addition, according to the Xie et al. (2019) [49], zinc ions have a certain antibacterial effect, which can more effectively prevent mold from attacking the adhesive. In order to verify the anti-mildew effect of zinc ions, the control experiments of the mBM-modified soybean meal-based adhesive (control 1) and the Zn-modified soybean meal-based adhesive (control 2) were conducted simultaneously for anti-mold test. It was found that the control 1 was mildewed only two days later, while the control 2 was not eroded by mold during the two-week anti-mold test (Appendix A). These results implied that the zinc ion played the roles of both cross-linking and anti-mildew in the adhesive system.

## 4. Conclusions

In summary, a soybean meal and blood meal composite bio-adhesive with water resistance, toughness, and mold resistance was developed. The wet shear strengths of all ion-modified adhesives were higher than 0.7 MPa. Especially, compared to the adhesive I, the wet shear strength of the ion-modified soybean meal blood meal-based adhesive was improved from 0.48 to 1.1 MPa, the solid content was increased from 30.9% to 34.3%, and the moisture uptake was decreased from 14.6% to 11.9%. These attractive properties could be attributed to the metal coordination between metal ions (Ca^2+^, Zn^2+^) and protein active groups, as well as the co-effect of blood meal. Moreover, the adhesive had excellent mold resistance without bringing in additional anti-mildew agent. In this study, meal, the processing residues of agricultural and sideline industries, were effectively utilized, which would greatly increase the value of these raw materials and expand their application scope.

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
