# Peer review of "A Tough and Mildew-Proof Soybean-Based Adhesive Inspired by Mussel and Algae"

_polymers, 2020, doi:10.3390/polym12040756_

Round 1

Reviewer 1 Report

The article, describing a soybean meal and blood meal adhesive. the work is comprehensive, with plenty of supporting evidence for the author's claims.

The title seems misleading. while metal coordination may be in place, the lack of description of use of BM and SM in the title gives the reader the wrong idea.

the authors uses superlatives ("outstanding performances"), these should be avoided.

The introduction is far from being thorough enough. Evidence of metal ion complexation in BM/SM wood adhesives exists elsewhere and are not mentioned (e.g Handbook of Adhesive Technology, by Pizzi and Mittal). This in turn may lead to the novelty of the work presented here being vague. 

The rational behind the study design is unclear, especially in how were the reference materials chosen. Since there are plenty of publications regarding BM/SM adhesives, the use of a known and published formulation as a reference would be preferable (at least in terms of  mechanical properties). This is even further relevant in light of the Author's previous work, where the shear strength values reached more than 1.2 MPa

Reviewer 2 Report

The manuscript reports about the formulation of a bio-based adhesive comprising soybean and blood meals. I think that the overall content and results are interesting. The manuscript is fairly well written and illustrated. I would recommend the journal publication, however, I would like the authors to address the comments below.

The discussion around Fig. 6 is a bit weak in my view. First of all, I would not mention about “toughness”. It is energy required for unit surface of crack propagation, but as it stand there are no evidence concerning neither absorbed energy nor crack length. This reviewer also feels that Fig. 6 does not really show any crack. I think this part should be improved.

Please add error bars in Fig. 7 concerning shear strength of the adhesive. It is important to highlight data reliability since one of the main conclusions of the manuscript is rooted on these results.

Reviewer 3 Report

The manuscript by Bai et al describes a protein-based adhesive system consisting of soybean meal and blood meal crosslinked by the diglycidyl ether of 1,6 hexanediol and co-ordinated with transition metal ions, characterization of formulated adhesive, and its potential application. The research work is interesting to the readers/researchers in protein-based adhesive. The manuscript may be accepted after minor revision and English polishing.  Below are some of the comments/suggestion.

  1. Abstract: Authors write “…..reaction of 1,6-hexane dioldiglycidyl ether with protein to form a preliminary cross-linking network; subsequently, the coordination of zinc ions with amino or carboxyl strengthened and toughened the adhesive. Finally, the calcium ions gelled the adhesives provided cohesion force and made the network structure more compact”. It would be appropriate to add some information about the components of adhesive formulation and/or how the adhesive formulation was developed.
  2. Abbreviations: the term blood meal and soybean meal are not that long terms that needs to be replaced with two-letter abbreviations. Instead of BM, it is recommended to write blood meal throughout the manuscript, and similarly soybean meal instead of SM.
  3. Line 30, authors write “due to the shortage of the petroleum resources” which is not really the case as of now. Rephrasing as “depleting petroleum resources” would be appropriate.
  4. Line 43 to 46, what is the term “composites” referring to?
  5. Lines 38 to 40, authors argue that the methods of developing protein-based adhesive by denaturation/crosslinking and biomimetic modification utilizes fossil resources. However, authors formulation consisted 1,6-hexane dioldiglycidyl ether, which is also a synthetic compound. What is the novelty of this work?
  6. Line 86, as BM is not soluble in water, a term “suspended” will be appropriate instead of dissolved.
  7. Adhesive names: the names “SM/mBM/HDE/Zn-1”, “SM/mBM/HDE/Zn-1.5”, or “SM/mBM/HDE/Zn-2” are not appropriate. It would probably be more appropriate to use formulation 1, formulation 2, formulation 3…. etc.
  8. Table 1. Please add a column that illustrates the percentage weight of each component in the formulation.
  9. Line 101: 180 g/m2 of adhesive was coated on each veneer layer. Is the weight on dry weight basis or the weight of the wet formulation?
  10. To the reviewer’s understanding, after co-ordination with metal ions, shifting of IR bands of functional groups like carbonyl, amide, amine etc generally occurs towards lower wavenumber region, which was also observed in the case of shifing of IR band at 1233 cm-1 with appearance of new peak at 1062 cm-1. Looking into the IR, the reviewer does not agree that the band at 1515cm-1 was shifted to 1528 cm-1; it is simply the broadening of band, which likely is due to co-ordination/chelation with metal ions; but a reasonable explanation is needed for this behavior.
  11. Line 187-188, authors write the chemical (e.g., alkali, sodium silicate) treatment destroyed the structure of protein…. What is the purpose/role of sodium silicate addition? To the reviewer’s understanding, sodium silicate acts as a filler, and thus enhances the adhesive property.
  12. Figure 2b, there is no obvious shift in λmax of three samples. If a vertical line is drawn (as in IR spectra), all the samples likely have the same λmax value. Such results are not adding any supporting evidence on what authors are arguing.
  13. Was anti-mold test conducted based on any standard testing method?
